# The Expression Patterns of *BECN1*, *LAMP2*, and *PINK1* Genes in Colorectal Cancer Are Potentially Regulated by Micrornas and CpG Islands: An In Silico Study

**DOI:** 10.3390/jcm9124020

**Published:** 2020-12-12

**Authors:** Martyna Bednarczyk, Edyta Fatyga, Sylwia Dzięgielewska-Gęsiak, Dariusz Waniczek, Beniamin Grabarek, Nikola Zmarzły, Grażyna Janikowska, Małgorzata Muc-Wierzgoń

**Affiliations:** 1Department of Internal Medicine, Medical University of Silesia in Katowice, 40-055 Katowice, Poland; martyna.bednarczyk@outlook.com (M.B.); efatyga@sum.edu.pl (E.F.); sgesiak@sum.edu.pl (S.D.-G.); 2Department of Propedeutics Surgery, Chair of General, Colorectal and Polytrauma Surgery, Medical University of Silesia in Katowice, SHS in Katowice, 40-055 Katowice, Poland; dwaniczek@sum.edu.pl; 3Department of Histology, Cytophysiology and Embryology, Faculty of Medicine in Zabrze, University of Technology in Katowice, 40-055 Katowice, Poland; bgrabarek7@gmail.com (B.G.); nikola.zmarzly@gmail.com (N.Z.); 4Department of Analytical Chemistry, Medical University of Silesia in Katowice, 40-055 Katowice, Poland; grazynajanikowska@wp.pl

**Keywords:** autophagy, colorectal cancer, expression genes, microRNAs (miRNAs), CpG islands

## Abstract

Background: Autophagy plays a dual role of tumor suppression and tumor promotion in colorectal cancer. The study aimed to find those microRNAs (miRNAs) important in *BECN1*, *LAMP2*, and *PINK1* regulation and to determine the possible role of the epigenetic changes in examined colorectal cancer using an in silico approach. Methods: A total of 44 pairs of surgically removed tumors at clinical stages I‒IV and healthy samples (marginal tissues) from patients’ guts were analyzed. Analysis of the obtained results was conducted using the PL-Grid Infrastructure and Statistica 12.0 program. The miRNAs and CpG islands were estimated using the microrna.org database and MethPrimer program. Results: The autophagy-related genes were shown to be able to be regulated by miRNAs (*BECN1*—49 mRNA, *LAMP2*—62 mRNA, *PINK1*—6 mRNA). It was observed that promotion regions containing at least one CpG region were present in the sequence of each gene. Conclusions: The in silico analysis performed allowed us to determine the possible role of epigenetic mechanisms of regulation gene expression, which may be an interesting therapeutic target in the treatment of colorectal cancer.

## 1. Introduction

Colorectal cancer (CRC) is the second leading cause of cancer-related death among both men and women. Several factors are associated with increased risk for the disease, including diet, obesity, lack of physical activity, tobacco use, and alcohol use. On the other hand, it is believed that higher intake of dietary fiber, green leafy vegetables, folate, and calcium are protective against the development of CRC. Moreover, the chance of developing CRC increases markedly after age 50. In recent years, a significant amount of cancer research has focused on identifying novel therapies and therapeutic advances in colorectal cancer. Longer life expectancy is due to the approval of new molecular therapies that are administered in combination with various cytotoxic drugs. Thus far, many genes involved in different stages of tumor development (oncogenes) have been identified and undoubtedly more will be detected, the most important of which are APC, K-ras, DCC, DPC4, p53, TGFβ, and RII [1,2,3]. This study focused on genes involved in autophagy in CRC. Recent research provides new data regarding the complex mechanisms involved in colorectal carcinogenesis. Among these, autophagy plays a key role as it causes the progression of CRC [4].

Autophagy is a catabolic process responsible for the degradation and elimination of abnormal or damaged cellular proteins and organelles [4,5]. The process is characterized by the formation of autophagosomes, which absorb the undesirable content of the cytoplasm and then deliver it to the lysosomes, where they are degraded by hydrolase [6,7,8].

Depending on the catalyzed substrate or method of delivery to the lysosome, we can distinguish different types of autophagy: macrophagy, chaperone-dependent autophagy, microphagy, and selective autophagy (e.g., mitophagy) [9]. Autophagy dysregulation is probably one of the main factors affecting cancer development. On the basis of previous reports, we can assume that autophagy disorders promote the growth of abnormal tumor tissue and stimulate the progression of cancer [10,11,12].

The contribution of autophagy to CRC depends on the clinical stage, and numerous studies suggest that it may play a dual role in tumor development. On the one hand, it influences the progression of CRC; on the other hand, it can inhibit carcinogenesis [3].

The initiation and progression of cancer are controlled by genetic and epigenetic mechanisms. Previous studies have indicated the crucial role of epigenetic mechanisms in the initiation and development of cancers. This includes DNA methylation, histone modifications, nucleosome changes, and noncoding RNAs such as microRNA (miRNA). Alterations in epigenetic processes can lead to gene malfunctions, and hence tumor transformation. Recent discoveries in epigenetics have led to a better understanding of the mechanisms underlying carcinogenesis and have identified potential biomarkers for early disease detection and monitoring in cancer patients [13,14]. DNA methylation and gene silencing via noncoding RNA (miRNA) seem to have the highest potential for introducing new biomarkers for cancer diagnosis. miRNAs play a key role in the post-transcriptional regulation of gene expression. This mechanism is involved in many important physiological processes, including cell differentiation, proliferation, adhesion, and the regulation of signaling pathways. According to existing research, miRNAs also play a crucial role in tumor formation and metastasis. However, the influence of miRNAs and methylation in autophagy has not been explored in vivo in colon cancer [15,16,17,18].

This research is a continuation of earlier studies in which microarray analysis was used to analyze the expression profile of 435 miRNAs related to autophagy in colorectal cancer. Of the 50 statistically significant genes, the three most important that are involved in the regulation of autophagy were selected. Each participates in a different type of autophagy: macrophagy—*BECN1*, chaperone-dependent autophagy—*LAMP2*, and mitophagy—*PINK1*. Changes in the transcriptional activity of the *BECN1* indicated a decreased expression of this gene in all cancer clinical stages (CSI–CSIV). *LAMP2* expression was reduced only in CSI, while in CSII to CSIV, overexpression was observed, with the highest transcription activity in CSII [19]. For PINK1, reduced expression was seen in all stages of cancer.

The aim of this study was to determine, via in silico methods, miRNA’s importance in *BECN1*, *LAMP2*, and *PINK1* regulation and their possible role in the epigenetic changes in colorectal cancer. We chose the three most important genes involved in autophagy: *BECN1*, *LAMP2*, and *PINK1*. These are the main regulatory genes involved in this process. *BECN1* encodes a regulatory protein involved in macrophagy and participates in the formation of the autophagosome. Moreover, the level of *BECN1* is higher in cancer cells compared to normal cells. High *BECN1* expression is associated with a poor prognosis [20]. *LAMP2* participates in chaperone-mediated autophagy, supplying the substrate to the lysosome, while *PINK1* is involved in mitophagy. *PINK1* is intimately involved with mitochondrial quality control, identifying damaged mitochondria, and targeting specific mitochondria for degradation.

## 2. Materials and Methods

### 2.1. Participants

A total of 44 pairs of surgically removed tumors at clinical stages (CSI–CSIV) and healthy samples (marginal tissues) from patients’ guts were analyzed according to the guidelines published in the ninth edition of the Union for International Cancer Control/American Joint Committee on Cancer handbook. Healthy control tissue specimens were collected from an area 10 mm outside of the histologically negative margin. The study protocol was approved by the Bioethical Committee of the Medical University of Silesia (KNW/0022/KB1/42/14), and informed consent was obtained from all patients.

### 2.2. Methods

The first step was to isolate the total RNA. The tissue material was homogenized (Kinematics, AG, Bern, Switzerland), then the total RNA was isolated according to the manufacturer’s instructions using TRIzol reagent (Life Technologies, Carlsbad, CA, USA). Next, RNA was purified with a Qiagen RNeasy Mini Kit (Qiagen, Hilden, Germany) in combination with DNase I digestion. A Gene Quant II (Pharmacia Biotech, Uppsala, Sweden) spectrophotometer was used to quantify the RNA concentration on the basis of an absorbance of 260 nm. A Pharmacia Biotech spectrophotometer (version, company, city, state, country) was used to quantify the RNA concentration on the basis of an absorbance of 260 nm. The transcriptional activity of the genes was determined by the microarray technique (Affymetrix, Santa Clara, CA, USA) using a HG-U133A chip.

Analysis of the obtained results was conducted using the PL-Grid Infrastructure (http://www.plgrid.pl/) and Statistica (StatSoft, Warsaw, Poland) 12.0 program. In statistical analysis, the significance level was set at *p* < 0.05.

Next, on the basis of bioinformatic databases (www.microrna.org), the search employed the miRanda-miRSVR algorithm to identify potential microRNA (miRNA), which can regulate the expression of the *LAMP2*, *BECN1*, and *PINK1* genes. The analysis was based on “SEEDMATCH”, conservativity, free enthalpy, and the sequence availability of the selected miRNA molecule.

Next, on the basis of other bioinformatic databases (https://www.ncbi.nlm.nih.gov/, http://emboss.bioinformatics.nl/cgi-bin/emboss/geecee, http://www.urogene.org/cgi-bin/methprimer/methprimer.cgi), we conducted an in silico assessment of the effect of methylation on the expression of the previously analyzed genes *BECN1* (NM_001313998.1), *LAMP2* (NM_001122606.1), and *PINK1* (NM_032409.2). The assessment was based on the accession number of the reference gene sequence in the NCBI database (NCBI Reference Sequence). The MetPrimer program (plus CpG Island Prediction) was used to perform the analysis, where the sequence of interesting genes in the FASTA format was pasted into the empty space. Next, the options “Use CpG island prediction for primer selection?” and “Pick MSP primers” were selected. Standard values were marked, where “island size” was greater than 100 nucleotides, “observed/expected CpG ratio” was 0.60, and “percentage of G plus C” was 50.0. The last step was to read the quantity, size, and location of CpG islands in every single gene sequence to be analyzed.

## 3. Results

The first stage of the study was to select three genes out of 50 mRNAs related to autophagy that differentiate tumor tissue from normal intestine. Changes in the transcriptional activity of *BECN1*, *LAMP2*, and *PINK1* in colorectal cancer are illustrated in Figure 1.

The transcriptional activity of the selected genes in four stages of colorectal adenocarcinoma was compared to that in unaltered intestines (control). The expression of *BECN1* (participates in macrophagy) and *PINK1* (participates in mitophagy) was reduced at every stage of cancer development. *LAMP2* (participates in chaperone-dependent autophagy) was downregulated in CSI, while in CSII, CSIII, and CSIV, an increase in transcriptional activity was observed.

The following results illustrate Table 1 and Table 2. The in silico determination of miRNA molecules potentially regulating expression of *BECN1*, *LAMP2*, and *PINK1* genes was performed on the basis of the miRSVR score (cutoff ≤ −0.7; Table 1). It was found that *BECN1* is regulated by 49 miRNAs; each of them can affect its regulation, but only eight have a miRSVR score below −0.7. For *LAMP2*, out of a total of 62 miRNAs, only 19 were selected with a value of miRSVR below −0.7. *PINK1* expression was regulated by six miRNA molecules; however, only three of them met the criteria of the miRSVR.

The ratio between the number of miRNAs complying with the miRSVR prerequisite ≤ −0.7 and the number of molecules potentially regulating expression of the specific gene was, for the individual genes, *BECN1* (8/49), *LAMP2* (19/62), and *PINK1* (3/6). The highest impact probability was determined between *BECN1* and miR-30a, miR-30b, miR-30c, miR-30d, and miR-30e (from miRSVR = −1.3166 to miRSVR = −1.3149); *LAMP2* and miR-133a and miR-133b (miRSVR = −1.1919); and *PINK1* and miR-124 and miR-506 (miRSVR = −0.8744).

Next, the in silico analysis involved determination of the possible role of methylation in the regulation of genes related to autophagy, with the use of the bioinformatic database. For this purpose, the percentage of CG dinucleotides was determined for each gene, and the number, size, and position of CpG islands in the gene sequence were determined. On the basis of the data, we observed that one CpG island was present in the sequence of each gene.

For the analyzed genes, the lowest percentage of CG dinucleotides was observed in *LAMP2*, and the highest proportion of CG pairs in *PINK1*. Each examined gene had one CpG island, suggesting the participation of methylation in gene expression regulation. The size of this island, expressed in base pairs (bp), was different for each gene, as was the position in the nucleotide sequence. On the basis of this analysis, we found no relationship between the share of GC pairs in the sequence and the size of the CpG island.

## 4. Discussion

The initiation and progression of cancer is controlled by both genetic and epigenetic mechanisms [21]. Several miRNAs have been found to play important roles in the regulation of autophagy, hence influencing cancer development and progression [19]. Other miRNAs regulate the expression of physiological processes such as cell growth, proliferation, differentiation, apoptosis, migration, and angiogenesis [22,23,24,25]. Thus, their potential as therapeutic targets to be used in personalized therapy has been confirmed over the last decade [26,27,28]. To date, only two clinical trials have been conducted on the use of miRNAs in cancer treatment. The results point to the need for further improvements and to the limited knowledge on miRNAs’ mechanisms of action [29].

Autophagy may have a dual function in the progression of CRC. The autophagy inhibitors are able to inhibit autophagy, promote cell apoptosis, and increase patient sensitivity to chemotherapy. Some findings have revealed that the autophagy inducers (for example, rapamycin) can reduce the migration capacity of CRC cells. Some miRNAs influence autophagy during stressful conditions (e.g., chemotherapy, malnutrition, and hypoxia). Studies have shown that miRNAs can increase their sensitivity to chemotherapy by regulating the level of autophagy in CRC cells, which has potential therapeutic value. Some miRNAs are aberrantly expressed in CRC, but the mechanism is not completely clear. There are assumptions that it is related to autophagy, but more research is needed to explore this mechanism [20].

Currently, the most commonly used cancer treatment is chemotherapy, but its effectiveness is limited because of its toxicity to healthy cells. In addition, inhibition of apoptosis is often observed in tumor cells, which increases their resistance to chemotherapy, radiotherapy, and other treatments. The conducted research shows that the stress that accompanies oncological patients causes resistance to apoptosis, stimulating autophagy, which provides energy to the cells. Drugs using chemotherapy also induce autophagy, leading to cell death. This is probably associated with the maturation and degradation process. Hence, autophagy should be considered a new therapeutic target [20,30,31]. However, it is necessary to carry out molecular research in this direction. Drugs used up to now, inhibiting autophagy, act directly on lysosomes, e.g., chloroquine [32]. Therefore, in our research, we focused on the possibility of using methylation and miRNA as new therapeutic strategies to inhibit genes related to autophagy. One of the methylating agents that may be used in clinical trials is temozolomide (TMZ), which has thus far been used in the treatment of glioma. However, in future studies, we want to assess its impact on CRC [33].

The premise for our research was our previous studies, in which we analyzed the gene expression profile involved in autophagy using oligonucleotide microarrays. The results showed significant changes in the expression profile of genes differentiating cancer sections from the control. Therefore, the selected genes were checked as to whether the changes in expression were regulated by miRNA and/or related to methylation in the promoter regions [18,19]. In terms of the performed studies, tumor cells often have shown an altered DNA methylation pattern in comparison with normal cells [19].

The selected genes, *BECN1*, *LAMP2*, and *PINK1*, are crucial in autophagy. Thus, they were the targets for determining, in silico, the potential contribution of DNA methylation and miRNA to autophagy. *BECN1* primarily participates in the formation of autophagosomes. It serves as a substrate that binds other proteins to the structure of a pre-autophagosome [34]. *BECN1* is the first connection between autophagy and cancer [35]. In some types of tumors, there is a decrease in the expression of Beklin1 (cancer of the liver, lungs), which indicates that autophagy may be a mechanism that inhibits the development of cancer [36,37]. In addition, in tumors of the gastrointestinal tract, higher levels of BECN1 are observed in the first stages of the disease. The activity of this protein decreases in subsequent stages [34].

The *LAMP2* gene encodes a lysosomal membrane protein involved in chaperone-dependent autophagy; increasing levels of this gene’s expression are seen in the early stages of cancer [38]. The substrate to be degraded is bound to the chaperone in the cytosol, then the substrate‒chaperone complex moves towards the lysosome, where it is bound by the LAMP2 transmembrane receptor [38,39,40]. Chaperone-dependent autophagy disorders occur in various pathological states, including cancer [41]. One of the most common CMA anomalies is a disorder in the functioning of the translocation complex. Increased autophagy activity may lead to the progression of cancer, contributing to the survival and proliferation of cancer cells. It is also important to increase the level of LAMP2A [38].

Changes in *LAMP2* gene expression in carcinogenesis are dependent on the type of cancer and its histopathological differentiation. Increased *LAMP2* expression is observed in poorly differentiated human gastric adenocarcinoma, hepatocellular carcinoma, salivary adenoid cystic carcinoma, and invasive prostate carcinoma, as well as in patients with esophageal squamous cell carcinoma [42,43,44,45,46]. In all of these examples, the increase in *LAMP2* occurred in the early stages of the cancer and indicated a poor prognosis for the patient. However, its increase in breast cancer was in response to an acidic extracellular microenvironment [47,48,49,50,51]. Decreased or silenced expression of *LAMP2* leads to inactivation of autophagy [43].

*PINK1* encodes a mitochondrial membrane protein that is involved in the regulation of mitophagy. Undamaged polarized mitochondria are characterized by low levels of PINK1 protein, which is degraded in the transmembrane space in a *PARL* proteasome manner [52,53]. During depolarization, the concentration of *PINK1* increases, which indicates damage to the mitochondria. Thus, *PINK1* is considered a cellular polarization sensor of mitochondria [54].

In some cancers, demethylation of normally methylated genes occurs (global hypomethylation), which leads to the activation of a given gene (e.g., *LAMP2* in the present study). Such a situation applies, for example, to suppressor genes. Global hypomethylation is a relatively late event in the development of cancer, in contrast to hypermethylation, which is an early phenomenon, significantly preceding the occurrence of overt disease symptoms. This fact gives us an opportunity to develop new, early diagnosis of neoplastic changes [52,54].

In this study, we attempted to determine the potential contribution of DNA methylation to regulating expression by in silico analysis of genes. The results obtained indicate a possible implication of the mechanism in changes in the gene expression profile since, for each of them, on the basis of the reference sequence, we found no relationship between the percentage of GC dinucleotides in the sequence and the size of the CpG islands. Analyzing the results, we can assume reduced methylation in the promoter regions of *LAMP2* in cancer with respect to the control. *BECN1* is silenced in cancer tissue, which indicates that DNA methylation may also be responsible for this mechanism [55,56,57]. There are not many studies on the expression of *PINK1* in cancer. The only available data suggest that silencing of the mentioned gene is also observed in tumors [58,59,60,61], similar to our findings. It is necessary to discover the mechanisms responsible for this process.

The ability of miRNA to negatively regulate the transcriptional activity of genes was determined by the SVR parameter [18]. The higher the miRSVR parameter, the more influence the molecule has on the regulation of gene expression. Beclin 1 is regulated by 49 miRNAs, of which 8 potentially affect the regulation (the miRSVR score is above 0.7). Five miRNAs out of the eight selected belong to the large mir-30 miRNA family, and they are expressed in many tissues and cell types. In addition, they are involved in a large number of pathways, e.g., tumor suppression or oncogenesis, apoptosis, epithelial–mesenchymal transition, and autophagy [13,23]. *BECN1*, considered the main regulator of autophagy, has been identified as a direct target for miR-30a, mainly in tumors. Other studies have shown that miR-30a inhibits *BECN1* activity, thereby inactivating autophagy. Zhang et al. found *BECN1* transcriptional activity in colon cancer, suggesting a decreased rate of autophagy in this tumor; thus, we can consider miR-30 as a suppressor of autophagy [18,24,25,62]. For the *LAMP2* gene, of the 62 miRNAs, 19 have a value of miRSVR above 0.7, from which miR-145 should protect cells against cancer development [26]. *PINK1* is regulated by only three miRNAs: miR-340, miR-124, and miR-506. The last two (miR-124 and miR-506) have been described in detail in the literature and are involved in cell growth, proliferation, and metastasis. Chen et al. showed that miR-124 and miR-506 undergo reduced transcriptional activity in colon cancer compared to normal tissue. They are also considered tumor suppressors, inhibiting the proliferation, invasion, and migration of cancer cells. Additionally, miR-124 and miR-506 are involved in DNA methylation, leading to global DNA hypomethylation by reducing DNMT methyltransferase activity (DNMT3B and DNMT1) [27].

In silico analysis allows for a determination of the role of epigenetic mechanisms in the regulation of the expression of analyzed genes, which is the starting point for further research. Thanks to such analysis, it is possible to assess which miRNAs are promising targets for further research on the choice of molecular markers and new therapeutic strategies.

## 5. Conclusions

In silico analysis of epigenetic mechanisms involved in the regulation of gene expression associated with autophagy emphasizes their role in the regulation of the analyzed genes in colorectal cancer. Disruption of DNA methylation and miRNA molecules associated with the silencing of specific genes may be an interesting therapeutic target in the treatment of this cancer. In addition, the bioinformatics tools used are a promising method that can be used to develop new therapeutic strategies.

## Figures and Tables

**Figure 1 jcm-09-04020-f001:**
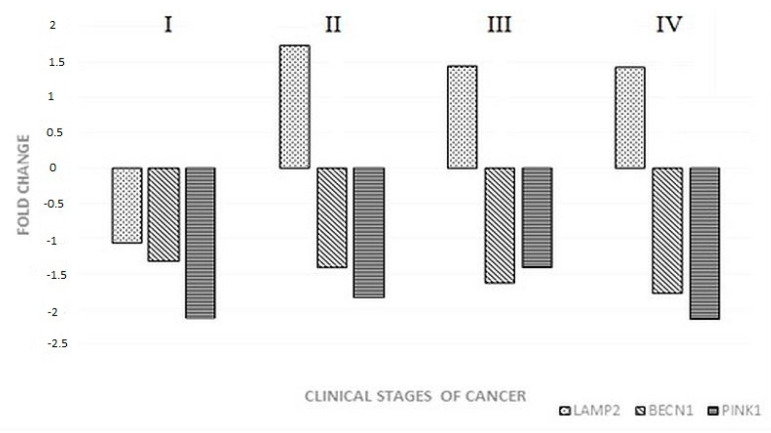
Changes in the expression of *LAMP2*, *BECN1*, and *PINK1* by clinical stage (I‒IV) of colorectal cancer.

**Table 1 jcm-09-04020-t001:** Number of miRNA differentiating control group (normal tissue) vs. research group (cancer tissue) transcriptomes; asymptotic *p*-value determined with Mann–Whitney nonparametric test.

FC	ID Number of miRNA	*p* < 0.05	*p* < 0.02	*p* < 0.01	*p* < 0.005	*p* < 0.001
FC > 1.0	435	11	2	0	0	0
FC > 1.1	258	11	2	0	0	0
FC > 1.5	43	1	1	0	0	0
FC > 2.0	11	0	0	0	0	0
FC > 3.0	3	0	0	0	0	0

FC: Fold Change.

**Table 2 jcm-09-04020-t002:** Autophagy-related ID of miRNA differentiating control group (normal tissue) vs. research groups (cancer tissue). Determined with nonparametric *t*-test with Benjamini‒Hochberg adjustment.

Probe Set ID	Gene Symbol	*p*-Value	Regulation	Fold Change
204892_x_at	EEF1A1	0.023226203	down	−1.3263694
208011_at	PTPN22	0.026081003	down	−1.1016078
209090_s_at	SH3GLB1	0.015157501	down	−1.7845742
210160_at	PAFAH1B2	0.024645526	down	−1.2792772
210670_at	PPY	0.030612411	up	1.3691329
213728_at	LAMP1	0.04813185	up	1.3584424
214636_at	CALCB	0.03949906	down	−1.1406626
215154_at	ULK2	0.04303478	down	−1.1541054
218214_at	ATG101	0.041259877	up	1.2474773
220405_at	SNTG1	0.01374268	down	−1.1522142
221774_x_at	SUPT20H	0.027980017	down	−1.2279038

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
