# Peer review of "The Expression Patterns of BECN1, LAMP2, and PINK1 Genes in Colorectal Cancer Are Potentially Regulated by Micrornas and CpG Islands: An In Silico Study"

_jcm, 2020, doi:10.3390/jcm9124020_

Round 1
Reviewer 1 Report
The article with the title “Expression pattern of BECN1, LAMP2 and PINK1 genes in colorectal cancer potentially regulated through microRNAs and CpG islands - in silico study” is in generally well done, but I would offer these comments to the investigators:
Introduction
- The manuscript focuses on the colorectal cancer but there is a lack of information about this type of cancer. I strongly recommend to add a paragraph with some information about CRC (epidemiology, Oncogenes and the role of autophagy in CRC development).
- Why you choose these three genes (BECN1, LAMP2 and PINK1) for your study. Please explain.
Discussion
- To date, only two clinical trials have been conducted with the use of miRNA in cancer treatment. Please provide the clinical numbers of the studies and if they have results please discuss them.
- Drugs using chemotherapy also induce autophagy, leading to cell death. Autophagy, as you also mentioned, has dual role. Several chemotherapeutic agents initiate or inhibit autophagy. Please revise.
- Therefore, in our research we focused on the possibility of using methylation and miRNA as new therapeutic strategies to inhibit genes related to autophagy. It will be useful to provide the name of methylating agents that may be used in clinical trials.
- Page 6: “Becline1” to “Beclin-1” Please revise.
References
- The vast majority of references are before 2014. Please provide latest references of the last 3 years where it is possible.
- Reference 48: Please provide the correct form of citation (journal, year, volume, number of pages).
Author Response
Dear Reviewer,
We highly appreciate the detailed valuable comments on Your review of our manuscript. The suggestions are quite helpful for us and we incorporate them in the revised paper.
Yours Sincerely
Prof. Małgorzata Muc-Wierzgoń
Response to comments:
- Why you choose these three genes (BECN1, LAMP2 and PINK1) for your study. Please explain.
I chose the three most important genes involved in autophagy: BECN1, LAMP2, PINK1.
These are the main regulatory genes involved in this process. BECN1 encodes a regulatory protein involved in macrophagy - participates in the formation of the autophagosome.
LAMP2, participates in chaperone-mediated autophagy - supplies the substrate to the lysosome.
While, PINK1 is involved in mitophagy. PINK1 is intimately involved with mitochondrial quality control by identifying damaged mitochondria and targeting specific mitochondria for degradation.
- To date, only two clinical trials have been conducted with the use of miRNA in cancer treatment. Please provide the clinical numbers of the studies and if they have results please discuss them.
NCT02369198 - A Phase I Study of TargomiRs as 2nd or 3rd Line Treatment for Patients With Recurrent MPM and NSCLC, Growth inhibition caused by miR-16 correlated with downregulation of target genes including Bcl-2 and CCND1, and miR-16 re-expression sensitised MPM cells to pemetrexed and gemcitabine.
NCT01829971 - A Multicenter Phase I Study of MRX34, MicroRNA miR-RX34 Liposomal Injection, Pharmacodynamic results showed delivery of miR-34a to tumours, and dose-dependent modulation of target gene expression in white blood cells
- Drugs using chemotherapy also induce autophagy, leading to cell death. Autophagy, as you also mentioned, has dual role. Several chemotherapeutic agents initiate or inhibit autophagy. Please revise.
Autophagy plays a dual role in carcinogenesis. Autophagy promotes cancer cell survival under stressful conditions or nutrient deprivation and thus may contribute to chemoresistance. The exaggerated and sustained autophagy that is trigged by anticancer therapies can lead to cell death in various cancers. Increased autophagy in the early stages of cancers can induce protection by suppressing tumorigenesis, necrosis, and chronic inflammation. Inhibition of autophagic influx may accelerate the initial steps of tumorigenesis and reduce protein degradation, and as a consequence, the reduced protein turnover might induce the early tumor progression.
In advanced stages, tumor cells use autophagy to survive cellular metabolic stress and to provide essential nutrients to tumor cells that are experiencing ischemia. Therefore, inhibiting autophagy in late-stage cancers can suppress tumor progression by blocking this prosurvival mechanism in nutrient-deprived tumor cells and by preventing protein recycling and cellular growth.
Autophagy facilitates cancer cell resistance to chemotherapy treatments, and the inhibition of autophagy may resensitize resistant tumor cells to anticancer therapy, thus enhancing the efficacy of the treatment.
- Therefore, in our research we focused on the possibility of using methylation and miRNA as new therapeutic strategies to inhibit genes related to autophagy. It will be useful to provide the name of methylating agents that may be used in clinical trials.
corrected
- Page 6: “Becline1” to “Beclin-1” Please revise.
corrected
- The vast majority of references are before 2014. Please provide latest references of the last 3 years where it is possible.
corrected
- Reference 48: Please provide the correct form of citation (journal, year, volume, number of pages).
corrected

Reviewer 2 Report
The authors do an in silico analysis on different parameters in colorectal cancer. Unfortunately there is very poor explanation on the significance of autophagy within this context and why they decided to focus on it. In addition, the authors do not provide a clear explanation on why they selected the three genes mentioned. Overall methodology is poorly explained and there are several mistakes within the text and references.
Lane 42. The kind of autophagy that targets specific organelles is not commonly referred to “specific autophagy” but rather “selective autophagy”.
Lane 42 and others. The expression “autophagy disorders” is rather confusing within this context, as it seems that it is referring to diseases arising from mutations in autophagy genes. I think the authors may consider changing this expression by autophagy dysregulation or autophagy modulation.
Lane 73-74. The reference to Table 1 seems to be wrong, since Table 1 contains a different information
Lane 105. Is it mRNA or miRNA?
Lane 129. Table 2 reference does not refer to the table 2 included in the paper. This must be clarified. What does it mean senior group within the table 2 legend?
Author Response
Dear Reviewer,
We highly appreciate the detailed valuable comments on Your review of our manuscript. The suggestions are quite helpful for us and we incorporate them in the revised paper.
Yours Sincerely
Prof. Małgorzata Muc-Wierzgoń
Response to comments:
The authors do an in silico analysis on different parameters in colorectal cancer. Unfortunately there is very poor explanation on the significance of autophagy within this context and why they decided to focus on it. In addition, the authors do not provide a clear explanation on why they selected the three genes mentioned. Overall methodology is poorly explained and there are several mistakes within the text and references.
A paragraph was added in the publication
Lane 42. The kind of autophagy that targets specific organelles is not commonly referred to “specific autophagy” but rather “selective autophagy”.
corrected
Lane 42 and others. The expression “autophagy disorders” is rather confusing within this context, as it seems that it is referring to diseases arising from mutations in autophagy genes. I think the authors may consider changing this expression by autophagy dysregulation or autophagy modulation.
Corrected
Lane 73-74. The reference to Table 1 seems to be wrong, since Table 1 contains a different information
Corrected
Lane 105. Is it mRNA or miRNA?
Here's the mRNA because the first step of the research was done on mRNA
Lane 129. Table 2 reference does not refer to the table 2 included in the paper. This must be clarified. What does it mean senior group within the table 2 legend?
Corrected

Round 2
Reviewer 1 Report
The authors answered in all issues.
Author Response
Dear Reviewer,
We highly appreciate the detailed valuable comments on Your review of our manuscript. The suggestions are quite helpful for us and we incorporate them in the revised paper.
English corrected as suggested by the MDPI English Editing System . The text has been checked for correct use of grammar and common technical terms, and edited to a level suitable for reporting research in a scholarly journal.
Yours Sincerely
Prof. Małgorzata Muc-Wierzgoń
Reviewer 2 Report
The authors made changes that improved the paper. However, there are a few issues to address:
- Lane 143. Reference to Table 2 seems to me wrong, should this be table 1?
- Table legends refer to “control and senior groups” in Table 1 and “control group (normal tissue) vs research groups (Cancer tissue)”. Are these the same groups? If so, they must be referred to in the same way. If they are not, it must be explained. This is very confusing for the reader.
Author Response
Dear Reviewer,
We highly appreciate the detailed valuable comments on Your review of our manuscript. The suggestions are quite helpful for us and we incorporate them in the revised paper.
Yours Sincerely
Prof. Małgorzata Muc-Wierzgoń
Response to comments:
English corrected as suggested by the MDPI English Editing System. The text has been checked for correct use of grammar and common technical terms, and edited to a level suitable for reporting research in a scholarly journal.
- Lane 143. Reference to Table 2 seems to me wrong, should this be table 1?
Reference was corrected, now it’s associated with Table 1
- Table legends refer to “control and senior groups” in Table 1 and “control group (normal tissue) vs research groups (Cancer tissue)”. Are these the same groups? If so, they must be referred to in the same way. If they are not, it must be explained. This is very confusing for the reader.
Table legends corrected, now they refer to in the same way.